# Fucose-Rich Sulfated Polysaccharides from Two Vietnamese Sea Cucumbers *Bohadschia argus* and *Holothuria (Theelothuria) spinifera*: Structures and Anticoagulant Activity

**DOI:** 10.3390/md20060380

**Published:** 2022-06-06

**Authors:** Nadezhda E. Ustyuzhanina, Maria I. Bilan, Andrey S. Dmitrenok, Eugenia A. Tsvetkova, Sofya P. Nikogosova, Cao Thi Thuy Hang, Pham Duc Thinh, Dinh Thanh Trung, Tran Thi Thanh Van, Alexander S. Shashkov, Anatolii I. Usov, Nikolay E. Nifantiev

**Affiliations:** 1The Laboratory of Glycoconjugate Chemistry, N.D. Zelinsky Institute of Organic Chemistry, Russian Academy of Sciences, Leninsky Prospect 47, 119991 Moscow, Russia; bilan@ioc.ac.ru (M.I.B.); dmt@ioc.ac.ru (A.S.D.); e_tsvet@ioc.ac.ru (E.A.T.); nextepwms@rambler.ru (S.P.N.); shash@ioc.ac.ru (A.S.S.); nen@ioc.ac.ru (N.E.N.); 2Chemical Analysis and Technology Development Department, NhaTrang Institute of Technology Research and Application, Vietnam Academy of Science and Technology, 02 Hung Vuong Street, Nhatrang 650000, Vietnam; caohang.nitra@gmail.com (C.T.T.H.); ducthinh.nitra@gmail.com (P.D.T.); dinthanhtrung410@gmail.com (D.T.T.); tranthanhvan@nitra.vast.vn (T.T.T.V.)

**Keywords:** sea cucumber, *Bohadschia argus*, *Holothuria (Theelothuria) spinifera*, fucosylated chondroitin sulfates, fucan sulfates, anticoagulant activity

## Abstract

Fucosylated chondroitin sulfates (FCSs) **FCS-BA** and **FCS-HS**, as well as fucan sulfates (FSs) **FS-BA-AT** and **FS-HS-AT** were isolated from the sea cucumbers *Bohadschia argus* and *Holothuria (Theelothuria) spinifera*, respectively. Purification of the polysaccharides was carried out by anion-exchange chromatography on DEAE-Sephacel column. Structural characterization of polysaccharides was performed in terms of monosaccharide and sulfate content, as well as using a series of non-destructive NMR spectroscopic methods. Both FCSs were shown to contain a chondroitin core [→3)-β-d-GalNAc-(1→4)-β-d-GlcA-(1→]_n_ bearing sulfated fucosyl branches at O-3 of every GlcA residue in the chain. These fucosyl residues were different in pattern of sulfation: **FCS-BA** contained Fuc2*S*4*S,* Fuc3S4S and Fuc4*S* at a ratio of 1:8:2, while **FCS-HS** contained these residues at a ratio of 2:2:1. Polysaccharides differed also in content of GalNAc4*S*6*S* and GalNAc4*S* units, the ratios being 14:1 for **FCS-BA** and 4:1 for **FCS-HS**. Both FCSs demonstrated significant anticoagulant activity in clotting time assay and potentiated inhibition of thrombin, but not of factor Xa. **FS-BA-AT** was shown to be a regular linear polymer of 4-linked α-L-fucopyranose 3-sulfate, the structure being confirmed by NMR spectra of desulfated polysaccharide. In spite of considerable sulfate content, **FS-BA-AT** was practically devoid of anticoagulant activity. **FS-HS-AT** cannot be purified completely from contamination of some FCS. Its structure was tentatively represented as a mixture of chains identical with **FS-BA-AT** and other chains built up of randomly sulfated alternating 4- and 3-linked α-L-fucopyranose residues.

## 1. Introduction

Two types of fucose-rich sulfated polysaccharides are known as components of marine invertebrates belonging to the class Holothuroidea (sea cucumbers). Unique fucosylated chondroitin sulfates (FCSs) have been found exclusively in the body walls of sea cucumbers. Molecules of these biopolymers have been shown to contain a linear core [→3)-β-d-GalNAc-(1→4)-β-d-GlcA-(1→]_n_ identical to the backbone of vertebrate chondroitin sulfates [1,2]. This chondroitin core usually contains α-l-fucosyl branches attached to O-3 of GlcA. Depending on the species of sea cucumber, FCSs may contain four types of GalNAc units (non-sulfated and sulfated at O-4, at O-6, or both at O-4 and O-6) [3], as well as GlcA, not only fucosylated at O-3, but also sulfated at O-3 or both at O-2 and O-3 [4,5]. In addition, disaccharide branches attached to O-3 of GlcA were found side by side with monofucosyl branches in several FCSs. Thus, FCS from *Holothuria* (*Ludwigothuria*) *grisea* was shown to contain the branch α-l-Fuc-(1→2)-α-l-Fuc3*S*-(1→ [6]. It should be noted that this structure was suggested after reinvestigation and correction of data described previously [7,8]. Similar difucoside branches were detected in other FCSs, examples are α-l-Fuc-(1→2)-α-l-Fuc3*S*4*S*-(1→ in FCS from *Eupentacta fraudatrix* [5], α-l-Fuc2S4S-(1→3)-α-l-Fuc4*S*-(1→ in FCS from *Stichopus japonicas* [3] and α-l-Fuc-(1→3)-α-l-Fuc4*S*-(1→ in FCS from *Holothuria lentiginosa* [9]. Recently more complex branches containing galactose or galactosamine residues have been found in several FCSs, examples are α-d-Gal4S(6S)-(1→2)-α-l-Fuc3*S*-(1→ in FCS from *Thelenota ananas* [10], α-d-GalNAc-(1→2)-α-l-Fuc3*S*4*S*-(1→ in FCS from *Acaudina molpadioides* [11] and α-d-GalNAcS-(1→2)-α-l-Fuc3*S*-(1→ in FCS from *Holothuria nobilis* [12]. There is some evidence that branches may be attached not only to O-3 of GlcA, but also to O-4 or O-6 of GalNAc residues of the backbone [13,14,15]. These examples demonstrate high structural diversity of FCSs. Since investigation of FCSs is connected with their promising biological activities [16,17,18,19], it is clear that different biological properties of FCS should depend on their fine structural characteristics, such as degree and position of sulfation, nature and position of branches and molecular mass distribution.

Another type of fucose-rich sulfated polysaccharides of sea cucumbers is represented by fucan sulfates (FSs), which are similar in many respects to FSs of sea urchins, but differ considerably from much more complex fucoidans of brown algae [20]. The simplest structures are polymers of 3- or 4-linked monosulfated α-l-fucose residues, examples being highly regular polysaccharides [-3)-α-l-Fuc2S-(1-]_n_ from *Stichopus horrens* [21,22] and *Stichopus herrmanni* [23]. Similar 3-linked FS, with a non-regular distribution of sulfate groups, was isolated from *Acaudina leicoprocta* [24], whereas 4-linked FSs, namely, [-4)-α-l-Fuc3S-(1-]_n_ and [-4)-α-l-Fuc2S-(1-]_n_, were found in *Holothuria fuscopunctata* and *Thelenota ananas*, respectively [22]. Structures of linear FSs are often represented as regular molecules built up of differently sulfated tetrasaccharide repeating units. Such polysaccharides, containing 3-linked backbone, were isolated from *Isostichopus badionotus* [25], *Acaudina molpadioides* [26], *Thelenota ananas* [27], *Pearsonothuria graeffei* [28], *Holothuria tubulosa* [29], *Holothuria polii* [30] and *Holothuria hilla* [31]. The structure of FS from *Holothuria albiventer* was shown to have a sulfated hexasaccharide repeating units [32], whereas sulfation pattern in FS from *Holothuria floridana* deviates considerably from the norm [33]. More complicated FSs contain other interfucoside linkages and/or branched carbohydrate chains. Thus, polysaccharides from *Holothuria edulis* and *Ludwigothurea grisea* contain backbones of 3-linked tetrafucosides connected by 1-2-linkages, where 2-substituted residues are additionally fucosylated (partially, as in *Ludwigothurea grisea*, or completely, as in *Holothuria edulis*) at position 4 [34]. Branched FS from *Apostichopus japonicus* contains a backbone of 3-linked α-l-Fuc2*S*, where every trisaccharide carries a non-sulfated 3-linked difucoside branch at-O-4 [35]. Recently some evidence has appeared on the possible presence of several structurally different FS in the same holothurian species. Thus, *Holothuria fuscopunctata* contains, in addition to 4-linked polymer of α-l-fucose 3-sulfate, another FS with tetrasaccharide repeating units -3)-α-l-Fuc2S,4S-(1-4)-α-l-Fuc-(1-3)-α-l-Fuc2S-(1-4)-α-l-Fuc-(1- with alternating 1-3 and 1-4 interfucoside linkages [36]. A new branched FS was isolated from *Pattalus mollis*. The polysaccharide contained a backbone of 4-linked α-l-Fuc2S, where every third residue was glycosylated at O-3 by α-l-Fuc4S or α-l-Fuc3S [37]. Further investigation of the same species made it possible to find, in addition to branched FS, the simultaneous presence of linear components, namely, randomly sulfated 3-linked fucan together with two differently sulfated 4-linked FS [38].

Natural fucose-enriched sulfated polysaccharides exerted excellent anticoagulant, antithrombotic, antivirus and anticancer activity together with many other biological actions [15,16,17,18,19]. Holothurian FCSs and FSs are isolated and characterized in order to find new biopolymers of practical importance and to establish distinct structure–activity correlations in these unique biologically active biopolymers. The Vietnamese coastal waters may be regarded as a rich source of marine invertebrates, including sea cucumbers, which contain both FCSs and FSs. For example, a novel FS with anticancer activity has been isolated from the Vietnamese sea cucumber *Stichopus variegatus* [39]. In the present communication we describe the isolation and structural characterization of sulfated polysaccharides from two Vietnamese holothurian species, *Bohadschia argus* and *Holothuria spinifera* (Appendix A). Some preliminary data on anticoagulant activity of these polysaccharides were obtained as well. It should be noted that FCS from *Bohadschia argus* has been carefully investigated previously [40], whereas accompanying FS and both FCS and FS of *Holothuria spinifera* have not been described in the literature. FCS isolated from *H. spinifera* was structurally similar to the corresponding FCS of *B. argus*. Both polysaccharides demonstrated anticoagulant activity *in vitro*, comparable with that of LMWH (enoxaparin). Surprisingly FS, isolated from *H. spinifera*, was practically inactive in these tests, in spite of its rather high sulfate content.

## 2. Results and Discussion

Crude extracts of sulfated polysaccharides were obtained from the body walls of sea cucumbers *Bohadschia argus* and *Holothuria spinifera* (Appendix A) by conventional solubilization of biomass in the presence of papain [8] followed by treatment of the extract with hexadecyltrimethylammonium bromide to precipitate the sulfated components, which were then transformed into water-soluble sodium salts by dissolving in 2 M NaCl and precipitation with ethanol, giving rise to crude sulfated polysaccharides **SP-BA** and **SP-HS**, respectively. Both crude extracts were subjected to anion-exchange chromatography on DEAE-Sephacel column. The fractions obtained as the result of chromatographic resolution are listed in Table 1.

Isolation of FCS from *Bohadschia argus* has been described previously, and the polysaccharide was used to obtain oligosaccharides acting as anticoagulants by intrinsic factor Xase complex inhibition [40]. According primarily to NMR spectral data, this polysaccharide had a typical FCS structure that was wholly 3-O-fucosylated GlcA and 4,6-disulfated GalNAc in the core with Fuc3S4S (~95%) and Fuc2S4S (~5%) as branches. In our work three fractions appeared (**FCS-BA1, FCS-BA2,** and **FCS-BA3**) by elution with water, 0.75 M and 1.0 M NaCl, respectively. These preparations were obtained in comparable yields of about 10% and contained GlcA, GalNAc, Fuc and sulfate in ratios near to the majority of known holothurian FCSs. Detection of minor Gal and GlcN in hydrolysates may be explained by the possible presence of small amounts of other GAGs, which could not be eliminated by anion-exchange chromatography. Appearance of a part of FCS eluted with water, and hence not absorbed on anion-exchanger, may probably be explained by the high molecular weight of **FCS-BA1** (cf. [38]), whereas **FCS-BA2** and **FCS-BA3,** having similar NMR spectra, were eluted separately, probably due to slightly different position of sulfation. All three FCS fractions had similar behavior in agarose gel electrophoresis (Appendix A). Fraction eluted with 1.0 M NaCl and designated as **FCS-BA** (Table 1) was used further for structural analysis.

The structure of **FCS-BA** was characterized using 1D and 2D NMR spectroscopy (Figure 1, Figure 2, Appendix A, Appendix A). The presence of Fuc, GalNAc and GlcA units in both polysaccharides was confirmed by the characteristic values of chemical shifts of C-6 for Fuc (δ 17.2 ppm) and GlcA (δ 176.0 ppm), as well as of C-2 for GalNAc (δ 52.7 ppm) in ^13^C NMR spectrum (Figure 1). The anomeric region in ^1^H NMR spectrum contained several low-field signals indicating the presence of different fucosyl branches (Figure 2). These spectra gave no doubts that **FCS-BA** belongs to the well-known class of holothurian FCSs [1,2,41,42]. It should be emphasized that very small intensity of **C**-6 signal in ^13^C NMR spectrum (δ 62.3 ppm, Figure 1) means that practically all the GalNAc residues are 4,6-disulfated. The ratio between residues **B** and **C** (Figure 3) was found to be 14:1. According to intensities of anomeric signals of sulfated Fuc residues it is possible to conclude that Fuc3S,4S (**E,** δ 5.34 ppm) predominates considerably over the two other branches (**D,** δ 5.68 ppm, and **F,** δ 5.40 ppm) in the polysaccharide molecule (Figure 3, the calculated ratio **D:E:F** = 1:8:2).

Chromatography of **SP-HS** gave rise to **FCS-HS**, which was eluted, as expected, with 1.0 M NaCl. Its electrophoretic mobility in agarose gel was identical to **FCS-BA** (Appendix A). According to NMR spectra (Figure 1 and Figure 2), which are very similar to those of **FCS-BA**, the polysaccharide undoubtedly belongs to FCSs. There are some minor structural differences between these two samples. Thus, the more intense signal at δ 62.3 ppm (Figure 1) corresponds to more substantial content of GalNAc residues non-sulfated at C-6 (the ratio between residues **B** and **C,** Figure 3, was found to be 4:1), whereas signals at δ 5.68 ppm and δ 5.34 ppm, having practically equal intensities (Figure 2), show the more substantial content of Fuc2S,4S in **FCS-HS**. The molar ratio between differently sulfated fucose residues **D:E:F** (Figure 3) was calculated as 2:2:1.

A rather unusual property of **SP-BA** is its incomplete solubility in water. The non-soluble gel-like fraction separated by centrifugation was treated with dilute acid in very mild conditions, giving rise to precipitate, which was shown to be mainly a protein. Neutralization of mother liquor followed by gel chromatography afforded **FS-BA-AT** (Table 1). Application of 1D and 2D NMR experiments (COSY, HSQC, and ROESY) led to assigning all the signals in NMR spectra (Figure 4 and Figure 5, Appendix A, Appendix A) and to establish the structure of **FS-BA-AT** as a regular linear polymer of 4-linked fucose 3-sulfate (Figure 6). As mentioned above, similar polysaccharide was isolated previously from *Holothuria fuscopunctata* [22]. The signal assignments presented in [22] and our data are given in Appendix A for comparison. The systematic differences up to 2 ppm in carbon chemical shifts between spectra of two polymers probably arise due to alternative conditions of signal registration used in these two works. The structure of linear backbone built up of 4-linked α-L-fucopyranose in **FS-BA-AT** was confirmed by NMR spectra of its desulfated preparation **FS-BA-AT-DS** (Appendix A).

Preparation **SP-HS** was also partially soluble in water, but in this case a rather small insoluble fraction (12%) was not investigated further. Anion-exchange chromatography afforded the main fraction **FS-HS**, not absorbed on the column and eluted with water. To improve the resolution of spectral signals, this fraction was treated with dilute acid, as above, to give **FS-HS-AT.** The NMR spectra of **FS-HS-AT** were shown to be rather complicated for detailed analysis due to overlapping of many important signals (Figure 4, Figure 5, Appendix A). This complexity may be explained by the random distribution of sulfates along the polymeric chains, since sulfation at every position causes a change in chemical shifts not only in its own residue, but also in both neighboring glycosylated and glycosylating residues. Nevertheless, the assignment of many signals and the corresponding correlations in the anomeric region of the spectra has been suggested (Appendix A) based on the assumption about the simultaneous presence of two types of polymers containing randomly sulfated repeating units shown in Figure 6. Further attempts to resolve the complex preparation **FS-HS-AT** are now in progress.

Since FCSs and FSs are known to demonstrate anticoagulant and antithrombotic activities [16,43], we have studied three isolated samples **FCS-BA, FCS-HS** and **FS-BA-AT** as anticoagulant agents in vitro. Low molecular weight heparin (enoxaparin) was used as standard (Figure 7). In the clotting time assay (APTT-test) the effects of branched **FCS-BA** and **FCS-HS** were higher than that of enoxaparin, while linear **FS-BA-AT** was almost inactive at the same concentrations (Figure 7A). The values of 2APTT (the concentrations that led to two-time increasing of clot formation) were 2.6 ± 0.1 μg/mL for **FCS-HS**, 3.1 ± 0.1 μg/mL for **FCS-BA**, and 3.8 ± 0.2 μg/mL for enoxaparin. The active samples **FCS-BA** and **FCS-HS** were studied further in the experiments with purified proteins thrombin and factor Xa (Figure 7B,C). It was shown that in the presence of anti-thrombin III (ATIII) **FCS-BA** and **FCS-HS** potentiate the inhibition of thrombin, although their effect was slightly lower than that of enoxaparin (Figure 7B). This activity may be explained by specific formation of the ternary complex between thrombin, antithrombin III and FCS. The possibility of such complexation was demonstrated by computer docking studies in one of our previous works [44]. At the same time both samples had very low activity against factor Xa (Figure 7C).

## 3. Materials and Methods

### 3.1. General Methods

Procedures for determination of neutral monosaccharides, amino sugars, uronic acids and sulfate content of polysaccharides have been described previously [45,46,47]. Solvolytic desulfation [45] was used to prepare **FS-HS-AT-DS** from **FS-HS-AT**. Molecular weights of polysaccharides were evaluated by chromatographic comparison with standard pullulans [48].

### 3.2. Isolation of Polysaccharides

Wild sea cucumbers *Holothuria (Theelothuria) spinifera* (Theel, 1886) and *Bohadschia argus* (Jaeger, 1833) with size range of 20–30 cm in length and 300–500 g in weight were collected in October 2020 at the coast-line of Nhatrang Bay, Vietnam. The samples were stored in sea water and transported to the laboratory in a day. After removing the viscera and washing, the fresh sea cucumber body walls were treated with 96% ethanol for 3 to 5 days, followed by soaking in acetone overnight at room temperature. Finally, the defatted samples were chopped and air-dried. Sulfated polysaccharides were isolated from the body wall of the sea cucumber using a method of Pham Duc Thinh et al., 2018 [39] (Figure 2). The defatted residue (50 g) was incubated with papain (10 g) in 1 L of 0.1 M sodium acetate buffer (pH 6) containing 5 mM EDTA and 5 mM cysteine at 60 °C for 24 h. The obtained mixture was heated at 100 °C for 20 min to inactivate the enzyme and to remove a small precipitate. The supernatant was treated with aqueous 10% hexadecyltrimethylammonium bromide solution (Cetavlon) up to complete precipitation of sulfated polysaccharides and left overnight at 4 °C. The resulting precipitate was centrifuged, washed with distilled water, dissolved in a mixture of 2 M NaCl and EtOH (4:1), precipitated with 3 volumes of 96% ethanol, and left at 4 °C for 24 h. The precipitate was centrifuged, washed with ethanol, dissolved in water, dialyzed, concentrated under a vacuum, and lyophilized to obtain crude sulfated polysaccharide **SP-HS** from *Holothuria spinifera* (yield 3,63%) and **SP-BA** from *Bohadschia argus* (yield 3,57%). Composition of crude polysaccharides is given in Table 1.

A suspension of 400 mg of **SP-BA** in 90 mL of water was stirred at room temperature for several hours and centrifuged at 6.000 rpm for 30 min. The gel-like precipitate was washed several times with water and lyophilized to give **FS-BA-ins**. The supernatant was placed on a column (3 × 10 cm) with DEAE-Sephacel (Saint Louis, MO, USA) in Cl¯-form and eluted with water followed by NaCl solution of increasing concentration (0.5, 0.75, 1.0 and 1.5 M), each time up to the absence of a positive reaction of eluate for carbohydrates [49]. Fractions were desalted on Sephadex G-15 (Saint Louis, MO, USA) column and lyophilized. Fractions eluted with water, 0.75 M NaCl and 1.0 M NaCl with the yields of 7.3, 11.8 and 11.8%, respectively, had similar composition and NMR spectra. The latter fraction was designated as **FCS-BA** and used further for structural analysis. Similar treatment of **SP-HS** gave rise to a rather small water-insoluble fraction (yield 12%), which was not investigated further. Chromatography of the water-soluble part of a sample on DEAE-Sephacel afforded the main fraction **FS-HS** eluted with water (yield 34.7%) and the second considerable fraction **FCS-HS** eluted with 1.0 M NaCl (yield 24.5%, Table 1).

### 3.3. Dilute Acid Treatment of Samples

A suspension of 170 mg of **FS-BA-ins** in 20 mL of 0.1 M HCl was stirred at 50 °C for 3 h, the insoluble material was centrifuged, washed and lyophilized to give a preparation (35.3%) containing 77.6% of protein and no carbohydrates. A viscous supernatant was neutralized by NaHCO_3_, desalted on Sephadex G-15 and chromatographed on DEAE-Sephacel, as above. The main polysaccharide fraction eluted with 1 M NaCl was desalted and lyophilized to afford **FS-BA-AT** (Table 1). Preparation **FS-HS-AT** was obtained after similar mild acid treatment of **FS-HS** (Table 1).

### 3.4. Agarose Gel Electrophoresis (PAGE)

Polysaccharides, heparin and enoxaparin (Clexan, Sanofi, Paris, France) (15 μg) were applied to 0.6% agarose gel prepared in 60 mM 1,3 diaminopropan-acetate buffer pH 9.0. Electrophoresis was run at 100 V in 60 mM 1,3 diaminopropan-acetate buffer pH 9.0 for 120 min. After migration gel was soaked in 0.1% cetyltrimethylammonium bromide for 30 min, dried, stained with 0.003% Stains-all in formamide-isopropanol-water (5:25:70) overnight in the dark and destained with water.

### 3.5. NMR Spectroscopy

The NMR spectra were recorded using the facilities of Zelinsky Institute Shared Center. Sample preparation and the conditions of experiments were described previously [50].

### 3.6. Anticoagulant Activity Measured in Clotting Time Test

The activated partial thromboplastin time assay for samples **FCS-BA**, **FCS-HS**, **FS-BA** and enoxaparin (Clexane^®^, Sanofi, Paris, France) was performed using Coatron^®^M2 coagulation analyzer (TECO, Munich, Germany) as described previously [4,51]. Briefly, a solution (10 µL) with concentration of 75 μg/mL, 37.5 μg/mL or 19.0 μg/mL of a polysaccharide sample (**FCS-BA**, **FCS-HS**, **FS-BA**, or enoxaparin) in purified water was added to 50 µL of normal plasma (Cormay, Poland). The mixture was incubated for 2 min at 37 °C, then 50 µL of APTT reagent (Cormay, Poland) was added, and the mixture was incubated again for 3 min at 37 °C. Then 50 µL of CaCl_2_ solution (0.025M) was added, and the time of clot formation was recorded. Purified water instead of a saccharide solution was used as a control.

### 3.7. Effect of Polysaccharides on Thrombin or Factor Xa Inactivation by Antithrombin III

Both experiments were carried out for samples **FCS-BA**, **FCS-HS**, and enoxaparin (Clexane^®^, Sanofi) at 37 °C in a 96-well plate using MultiscanGo (Thermo, Stockholm, Sweden) as described previously [15]. Briefly, a solution of polysaccharide sample (**FCS-BA**, **FCS-HS**, or enoxaparin) (20 µL) with concentrations of 500, 50, 5, 0.5, 0.05 μg/mL in Tris-HCl buffer (0.15 µM, pH 8.4) was added to 50 µL of a solution of ATIII (0.2 U/mL, Renam, Moscow, Russia) in Tris-HCl buffer. After 3 min incubation an aqueous solution of thrombin (50 µL, 20 U/mL, Renam, Russia) was added, and the mixture was incubated for 2 min. Then a chromogenic substrate (50 µL, 2 mM, Renam, Russia) was added, and the mixture was kept for 2 min. Absorbance of p-nitroaniline (405 nm) was measured.

To 50 µL of a solution of ATIII (0.5 U/mL) in Tris-HCl buffer a solution of a polysaccharide sample (**FCS-BA**, **FCS-HS** or enoxaparin) (20 µL) with concentrations of 500, 50, 5, 0.5, 0.05 μg/mL in Tris-HCl buffer was added. After 3 min incubation an aqueous solution of factor Xa (50 µL, 2 U/mL, Renam, Russia) was added, and the mixture was incubated for 2 min. Then a chromogenic substrate (50 µL, 2 mM, Renam, Russia) was added, and the mixture was kept for 2 min. Absorbance of p-nitroaniline (405 nm) was measured.

### 3.8. Statistical Analysis

All biological experiments were performed in quadruplicate (n = 4). The results are presented as Mean ± SD. Statistical significance was determined with Student’s *t* test. The *p* values less than 0.05 were considered as significant.

## 4. Conclusions

Fucosylated chondroitin sulfate **FCS-BA** and fucan sulfate **FS-BA** were isolated from the sea cucumber *Bohadschia argus.* Similar preparations **FCS-HS** and **FS-HS** were obtained from *Holothuria (Theelothuria) spinifera*. The main components of both FCSs were GlcA, GalNAc, Fuc and sulfate. Based on the data of NMR spectroscopy, both polysaccharides were shown to contain chondroitin core [→3)-β-d-GalNAc-(1→4)-β-d-GlcA-(1→]_n_ bearing sulfated fucosyl branches at O-3 of every GlcA residue in the chain. These fucosyl residues were different in pattern of sulfation: **FCS-BA** contained Fuc2*S*4*S,* Fuc3S4S and Fuc4*S* in a ratio of 1:8:2, while **FCS-HS** included Fuc2*S*4*S*, Fuc3*S*4*S* and Fuc4*S* in a ratio of 2:2:1. Moreover, the polysaccharides differ also in GalNAc4*S*6*S* and GalNAc4*S* units content, the ratios being 14:1 for **FCS-BA** and 4:1 for **FCS-HS**. Both polysaccharides demonstrated significant anticoagulant activity in clotting time assay. This activity is probably connected with the ability of these FCSs to potentiate inhibition of thrombin by formation of ternary complex with thrombin and antithrombin III. Such complexation was confirmed previously by computer docking experiments [44]. At the same time activity of these FCSs as inhibitors of Xa was rather low. Fucan sulfate **FS-BA** was shown to be a linear polymer of 4-linked α-L-fucopyranose 3-sulfate, structure being confirmed by NMR spectra of the native polysaccharide and its desulfated derivative. It is interesting to mention that **FS-BA**, despite its rather substantial sulfate content, was practically devoid of anticoagulant activity. **FS-HS** had much more complex NMR spectra, and its structure was tentatively represented as a polysaccharide containing fragments which coincide with **FS-BA**, together with other fragments built up of randomly sulfated alternating 4- and 3-linked α-L-fucopyranose residues.

## Figures and Tables

**Figure 1 marinedrugs-20-00380-f001:**
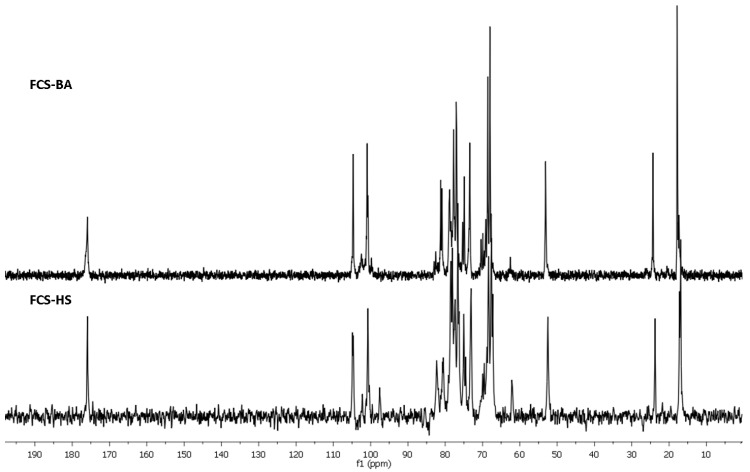
The ^13^C NMR spectra of fucosylated chondroitin sulfates **FCS-BA** and **FCS-HS**.

**Figure 2 marinedrugs-20-00380-f002:**
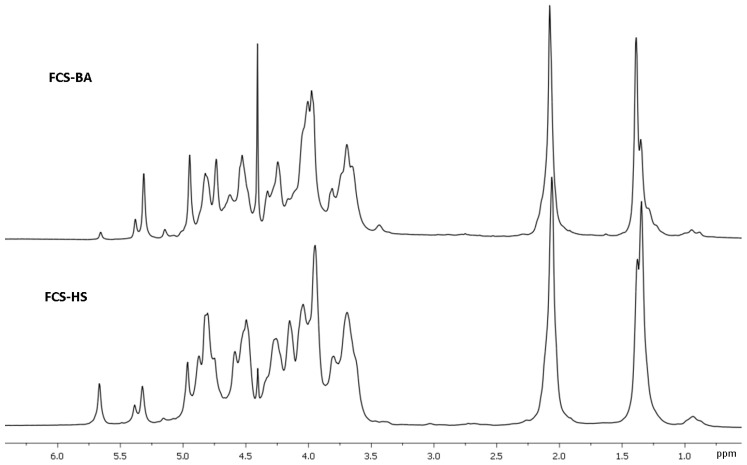
Fragments of ^1^H NMR spectra of fucosylated chondroitin sulfates **FCS-BA** and **FCS-HS**.

**Figure 3 marinedrugs-20-00380-f003:**
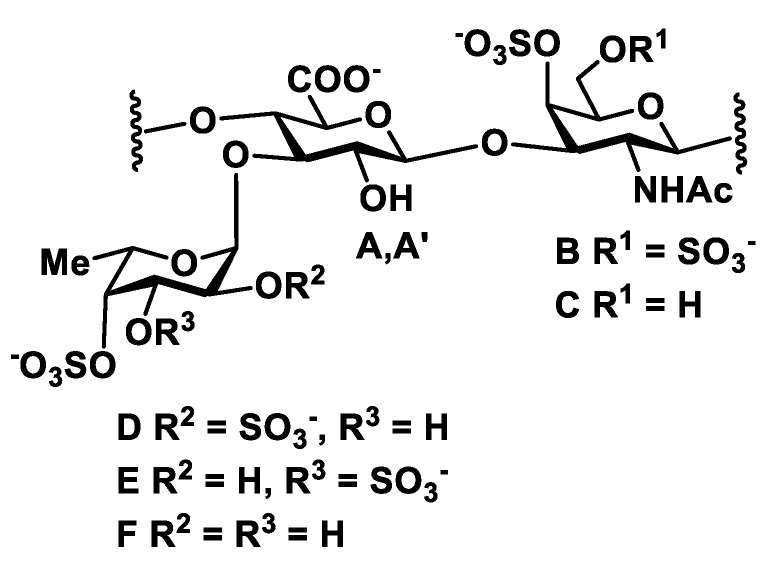
Repeating blocks of fucosylated chondroitin sulfates **FCS-BA** and **FCS-HS**. Unit **A** bears Fuc2*S*4*S* (**D**), whereas unit **A’** bears Fuc3*S*4*S* (**E**) or Fuc4*S* (**F**).

**Figure 4 marinedrugs-20-00380-f004:**
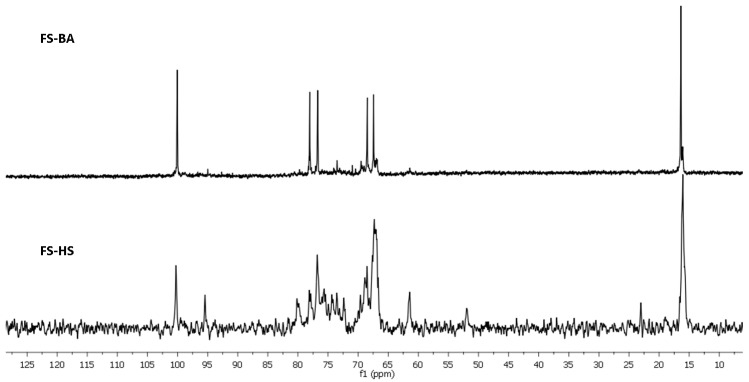
The ^13^C NMR spectra of sulfated fucans **FS-BA** and **FS-HS**.

**Figure 5 marinedrugs-20-00380-f005:**
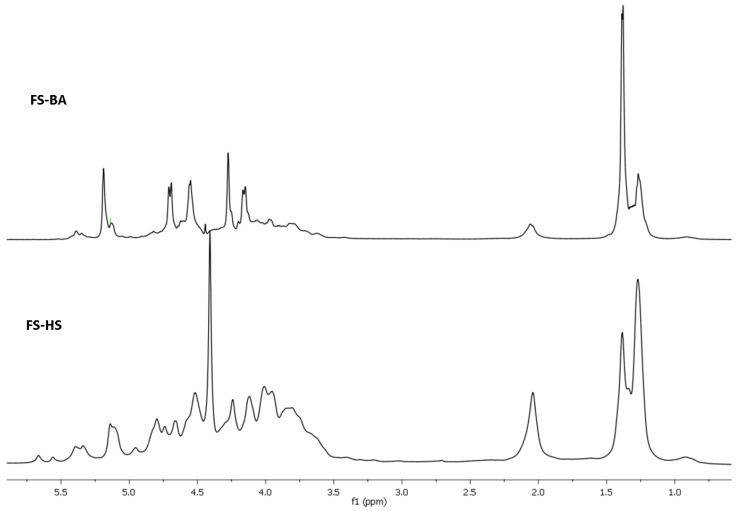
The ^1^H NMR spectra of sulfated fucans **FS-BA** and **FS-HS**.

**Figure 6 marinedrugs-20-00380-f006:**
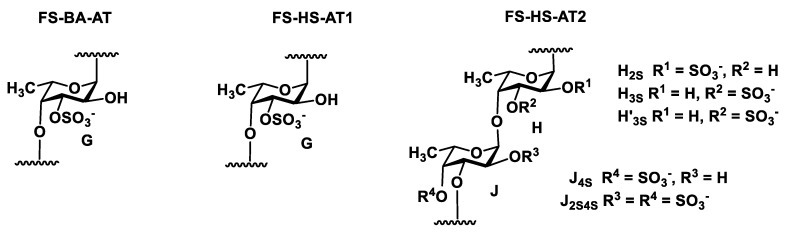
Repeating blocks of fucans **FS-BA-AT** and **FS-HS-AT**.

**Figure 7 marinedrugs-20-00380-f007:**
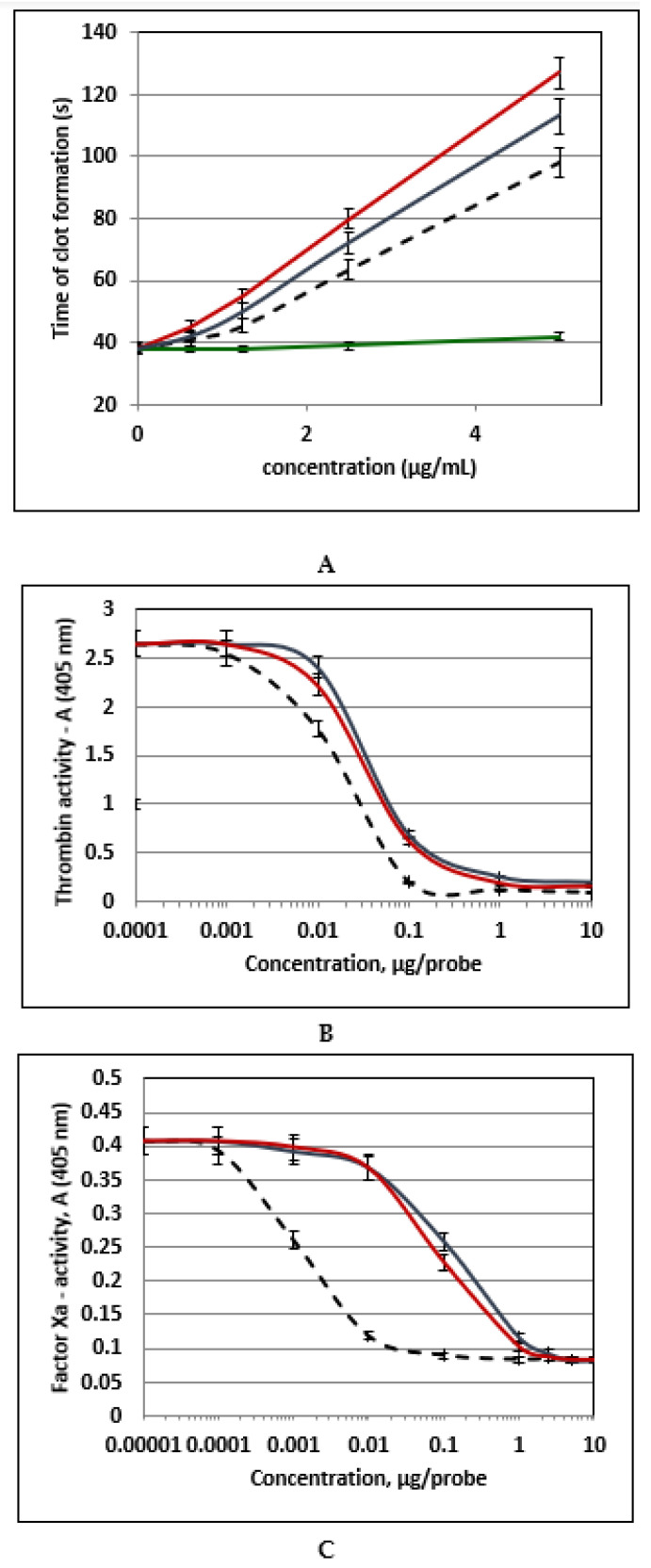
Anticoagulant activity of polysaccharides **FCS-BA** (blue), **FCS-HS** (red), **FS-BA-AT** (green) and enoxaparin (dotted line). (**A**) APTT assay, (**B**) Anti-IIa-activity in the presence of ATIII, (**C**) Anti-Xa-activity in the presence of ATIII. *n* = 4, *p* < 0.05.

**Table 1 marinedrugs-20-00380-t001:** Characteristics of crude polysaccharide preparations **SP-BA and SP-HS** and the fractions obtained by their chromatography on DEAE-Sephacel and then used for structural analysis (composition in molar ratios relative to fucose).

Sample	Fuc	Gal	GlcNAc	GalNAc	UA, Na-salt	SO_3_Na	Molecular Weight, kDa (Dispercity)
**SP-BA**	1.00	0.14	0.07	0.30	0.30	2.05	
**FCS-BA**	1.00	0.09	n.d.	0.78	0.83	4.04	32 (1.55)
**FS-BA-AT**	1.00	0.09	0.04	0.06	0.04	1.28	55 (1.35)
**SP-HS**	1.00	0.12	0.08	0.3	0.21	2.44	
**FCS-HS**	1.00	0.12	0.04	0.74	1.10	3.44	30 (1.62)
**FS-HS-AT**	1.00	0.09	0.04	0.18	0.18	1.69

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
