# Peer review of "Fucose-Rich Sulfated Polysaccharides from Two Vietnamese Sea Cucumbers Bohadschia argus and Holothuria (Theelothuria) spinifera: Structures and Anticoagulant Activity"

_marinedrugs, 2022, doi:10.3390/md20060380_

Round 1

Reviewer 1 Report

Dear Editor,

Thank you very much for the opportunity to review this interesting article.

The article “Fucose-reach sulfated polysaccharides from two Vietnamese sea cucumbers Bohadschia argus and Holothuria (Theelothuria) spinifera: structures and anticoagulant activity” by Nadezhda E. Ustyuzhanina and Collegues report the NMR-structural characterization of two typologies of acidic polysaccharides, fucosylated chondroitin sulfates and fucan sulfates, extracted by papain treatment and purified by two sequential precipitation steps followed by anion exchange chromatography, from two species of sea cucumbers harvested from the coast-line of Nhatrang Bay, Vietnam. Furthermore, their anticoagulant activities were assessed.

The manuscript is clear enough and results are quite interesting.

However, moderate changes should be done before publication:

-        please, carefully proof-read the manuscript to minimize typing errors. Hereafter, some of them are reported as examples:

§  row 2: Fucose-reach should be replaced with Fucose-rich;

§  row 134: 1.0M should be replaced with 1.0 M;

§  row 206: in should be replaced with at:

§  row 270: during should be replaced with for.

-        For the sake of clarity, I would like to ask Authors to change the figure S2, which is a scheme describing the main procedures for isolating the crude sulfated polysaccharides from the two sea cucumber species, with a less detailed workflow reporting instead the whole procedures and the different fractions obtained, together with their identity.

-        Moreover, the legend to figure S3 should be implemented. In the actual form is quite unclear.

Overall, I have no major concerns about the publication of this paper.

Best regards,

Author Response

We would like to express our great appreciation for your comments and  suggestions concerning our manuscript. The text of the paper has been modified accordingly, and the detailed corrections have been displayed in red in the revised manuscript.

-        please, carefully proof-read the manuscript to minimize typing errors. Hereafter, some of them are reported as examples:

We apologize for the grammatical errors and have revised the whole manuscript according to your suggestion.

  • row 2: Fucose-reach should be replaced with Fucose-rich; corrected.
  • row 134: 1.0M should be replaced with 1.0 M; - corrected.
  • row 206: in should be replaced with at: corrected.
  • row 270: during should be replaced with for.- corrected.

-        For the sake of clarity, I would like to ask Authors to change the figure S2, which is a scheme describing the main procedures for isolating the crude sulfated polysaccharides from the two sea cucumber species, with a less detailed workflow reporting instead the whole procedures and the different fractions obtained, together with their identity.

Since figure S2 represents traditional scheme used many times for isolation of crude sulfated polysaccharides from holothurian body walls, we suggest to exclude it from the Supplementary materials. Accordingly Figure S3 should be designated as S2. Further fractionation procedures of crude polysaccharides are different for two different sea cucumbers, so it is rather difficult to represent them in one scheme.

-        Moreover, the legend to figure S3 should be implemented. In the actual form is quite unclear.

The legend to figure S3 is enlarged by decoding the abbreviations.

Reviewer 2 Report

Title: Fucose-reach sulfated polysaccharides from two Vietnamese sea cucumbers Bohadschia argus and Holothuria (Theelothuria) spinifera: structures and anticoagulant activity

Herein, the authors aimed to describe the isolation and structural characterization of sulfated polysaccharides from two Vietnamese holothurian species, Bohadschia argus and Holothuria spinifera .

Major point

Mechanistic in vitro studies should be performed to trace the mode of action as anticoagulant agent or in silico studies employing both molecular docking and molecular dynamics simulation

Minor points

·         What is new should be highlighted more comprehensively in the introduction section

·         There are  some spelling and grammatical that should be revised carefully

Author Response

We would like to express our great appreciation for your comments and suggestions concerning our manuscript. The text of the paper has been modified accordingly, and the detailed corrections have been displayed in red in the revised manuscript.

Mechanistic in vitro studies should be performed to trace the mode of action as anticoagulant agent or in silico studies employing both molecular docking and molecular dynamics simulation

Computational study of the possible formation of the ternary complex between thrombin, antithrombin III and fucosylated chondroitin sulfates was the subject of one of our previous investigation. The results are now mentioned both in Results and Discussion and in Conclusion. The corresponding reference is added into the Reference list.

The text now added to Results and Discussion:

This activity may be explained by specific formation of the ternary complex between thrombin, antithrombin III and FCS. The possibility of such complexation was demonstrated by computer docking studies in one of our previous works [44].

The text now added to Conclusion:

This activity is probably connected with the ability of these FCSs to potentiate inhibition of thrombin by formation of ternary complex with thrombin and antithrombin III. Such complexation was confirmed previously by computer docking experiments [44].

Minor points

  • What is new should be highlighted more comprehensively in the introduction section

The text is now added to the Introduction:

FCS isolated from H. spinifera was structurally similar to the corresponding FCS of B. argus. Both polysaccharides demonstrated anticoagulant activity in vitro, comparable with that of LMWH (enoxaparin). Surprisingly FS, isolated from H. spinifera, was practically inactive in these tests, in spite of its rather high sulfate content.

  • There are some spelling and grammatical errors that should be revised carefully.

We apologize for the grammatical errors and have revised the whole manuscript according to your suggestion.

Reviewer 3 Report

This manuscript has revealed the structural details in fucose-rich sulfated polysaccharides and their anticoagulant activity. The chemical analysis was performed properly. The structural details were well related with functional activities. This manuscript can be published after minor text editing (ex. reach -> rich in the title). 

Author Response

We would like to express our great appreciation for your comments and suggestions concerning our manuscript. The text of the paper has been modified accordingly, and the detailed corrections have been displayed in red in the revised manuscript.

We apologize for the grammatical errors and have revised the whole manuscript according to your suggestion.

Round 2

Reviewer 2 Report

No additional comments